# Reducing Network Agnostophobia

**Akshay Raj Dhamija, Manuel Günther, and Terrance E. Boult**
Vision and Security Technology Lab, University of Colorado Colorado Springs
{adhamija | mgunther | tboult} @ vast.uccs.edu

## Abstract

Agnostophobia, the fear of the unknown, can be experienced by deep learning engineers while applying their networks to real-world applications. Unfortunately, network behavior is not well defined for inputs far from a networks training set. In an uncontrolled environment, networks face many instances that are not of interest to them and have to be rejected in order to avoid a false positive. This problem has previously been tackled by researchers by either *a*) thresholding softmax, which by construction cannot return *none of the known classes*, or *b*) using an additional background or garbage class. In this paper, we show that both of these approaches help, but are generally insufficient when previously unseen classes are encountered. We also introduce a new evaluation metric that focuses on comparing the performance of multiple approaches in scenarios where such unseen classes or unknowns are encountered. Our major contributions are simple yet effective Entropic Open-Set and Objectosphere losses that train networks using negative samples from some classes. These novel losses are designed to maximize entropy for unknown inputs while increasing separation in deep feature space by modifying magnitudes of known and unknown samples. Experiments on networks trained to classify classes from MNIST and CIFAR-10 show that our novel loss functions are significantly better at dealing with unknown inputs from datasets such as Devanagari, NotMNIST, CIFAR-100, and SVHN.

## 1 Introduction and Problem Formulation

Ever since a convolutional neural network (CNN) [19] won the ImageNet Large Scale Visual Recognition Challenge (ILSVRC) in 2012 [33], the extraordinary increase in the performance of deep learning architectures has contributed to the growing application of computer vision algorithms. Many of these algorithms presume detection before classification or directly belong to the category of detection algorithms, ranging from object detection [13, 12, 32, 23, 31], face detection [17], pedestrian detection [42] etc. Interestingly, though each year new state-of-the-art-algorithms emerge from each of these domains, a crucial component of their architecture remains unchanged – handling unwanted or unknown inputs.

Object detectors have evolved over time from using feature-based detectors to sliding windows [34], region proposals [32], and, finally, to anchor boxes [31]. The majority of these approaches can be seen as having two parts, the proposal network and the classification network. During training, the classification network includes a background class to identify a proposal as not having an object of interest. However, even for the state-of-the-art systems it has been reported that the object proposals to the classifier "*still contain a large proportion of background regions*" and "*the existence of many background samples makes the feature representation capture less intra-category variance and more inter-category variance (...) causing many false positives between ambiguous object categories*" [41].

In a system that both detects and recognizes objects, the ability to handle unknown samples is crucial. Our goal is to improve the ability to classify correct classes while reducing the impact of unknown inputs. In order to better understand the problem, let us assume $\mathcal{Y} \subset \mathbb{N}$ be the infinite label space of all classes, which can be broadly categorized into:

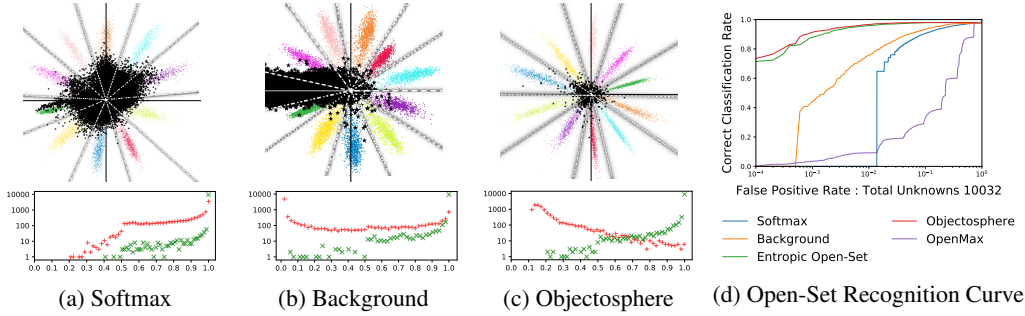

| (a) Softmax | (b) Background | (c) Objectosphere | (d) Open-Set Recognition Curve |

Figure 1: LeNet++ Responses To Knowns and Unknowns. *The network in (a) was only trained to classify the 10 MNIST classes ($\mathcal{D}'_c$) using softmax, while the networks in (b) and (c) added NIST letters [15] as known unknowns ($\mathcal{D}'_b$) trained with softmax or our novel Objectosphere loss. In the feature representation plots on top, colored dots represent test samples from the ten MNIST classes ($\mathcal{D}_c$), while black dots represent samples from the Devanagari[28] dataset ($\mathcal{D}_a$), and the dashed gray-white lines indicate class borders where softmax scores for neighboring classes are equal. This paper addresses how to improve recognition by reducing the overlap of network features from known samples $\mathcal{D}_c$ with features from unknown samples $\mathcal{D}_u$. The figures in the bottom are histograms of softmax probability values for samples of $\mathcal{D}_c$ and $\mathcal{D}_a$ with a logarithmic vertical axis. For known samples $\mathcal{D}_c$, the probability of the correct class is used, while for samples of $\mathcal{D}_a$ the maximum probability of any known class is displayed. In an application, a score threshold $\theta$ should be chosen to optimally separate unknown from known samples. Unfortunately, such a threshold is difficult to find for either (a) or (b), a better separation is achievable with the Objectosphere loss (c). The proposed Open-Set Classification Rate (OSCR) curve in (d) depicts the high accuracy of our approach even at a low false positive rate.*

- $\mathcal{C} = \{1, \ldots, C\} \subset \mathcal{Y}$: The *known classes of interest* that the network shall identify.
- $\mathcal{U} = \mathcal{Y} \setminus \mathcal{C}$: The *unknown classes* containing all types of classes the network needs to reject. Since $\mathcal{Y}$ is infinite and $\mathcal{C}$ is finite, $\mathcal{U}$ is also infinite. The set $\mathcal{U}$ can further be divided:
  1. $\mathcal{B} \subset \mathcal{U}$: The *background*, *garbage*, or *known unknown* classes. Since $\mathcal{U}$ is infinitely large, during training only a small subset $\mathcal{B}$ can be used.
  2. $\mathcal{A} = \mathcal{U} \setminus \mathcal{B} = \mathcal{Y} \setminus (\mathcal{C} \cup \mathcal{B})$: The *unknown unknown* classes, which represent the rest of the infinite space $\mathcal{U}$, samples from which are not available during training, but only occur at test time.

Let the samples seen during training belonging to $\mathcal{B}$ be depicted as $\mathcal{D}'_b$ and the ones seen during testing depicted as $\mathcal{D}_b$. Similarly, the samples seen during testing belonging to $\mathcal{A}$ are represented as $\mathcal{D}_a$. The samples belonging to the known classes of interest $\mathcal{C}$, seen during training and testing are represented as $\mathcal{D}'_c$ and $\mathcal{D}_c$, respectively. Finally, we call the unknown test samples $\mathcal{D}_u = \mathcal{D}_b \cup \mathcal{D}_a$.

In this paper, we introduce two novel loss functions that do not directly focus on rejecting unknowns, but on developing deep features that are more robust to unknown inputs. When training our models with samples from the background $\mathcal{D}'_b$, we do not add an additional softmax output for the background class. Instead, for $x \in \mathcal{D}'_b$, the Entropic Open-Set loss maximizes entropy at the softmax layer. The Objectosphere loss additionally reduces deep feature magnitude, which in turn minimizes the softmax responses of unknown samples. Both yield networks where thresholding on the softmax scores is effective at rejecting unknown samples $x \in \mathcal{D}_u$. Our approach is largely orthogonal to and could be integrated with multiple prior works such as [1, 11, 20], all of which build upon network outputs. The novel model of this paper may also be used to improve the performance of classification module of detection networks by better handling false positives from the region proposal network.

**Our Contributions:** In this paper, we make four major contributions: *a)* we derive a novel loss function, the Entropic Open-Set loss, which increases the entropy of the softmax scores for background training samples and improves the handling of background and unknown inputs, *b)* we extend that loss into the Objectosphere loss, which further increases softmax entropy and performance by minimizing the Euclidean length of deep representations of unknown samples, *c)* we propose a new evaluation metric for comparing the performance of different approaches under the presence of unknown samples, and *d)* we show that the new loss functions advance the state of the art for open-set image classification. Our code is publicly available.[1]

## 2 Background and Related Work

For traditional learning systems, learning with rejection or background classes has been around for decades [6], [5]. Recently, approaches for deep networks have been developed that more formally address the rejection of samples $x \in \mathcal{D}_u$. These approaches are called *open-set* [1, 4, 3], *outlier-rejection* [40, 25], *out-of-distribution detection* [38], or *selective prediction* [11]. In addition, there is also active research in network-based *uncertainty estimation* [14, 10, 36, 20].

In these prior works there are two goals. First, for a sample $x$ of class $\hat{c} \in \mathcal{C}$, $P(c \mid x)$ is computed such that $\arg \max_c P(c \mid x) = \hat{c}$. Second, for a sample $x$ of class $u \in \mathcal{U}$, either the system provides an uncertainty score $P(\mathcal{U} \mid x)$ or the system provides a low $P(c \mid x)$ from which $\arg \max_c P(c \mid x)$ is thresholded to reject a sample as unknown. Rather than approximating $P(u \mid x)$, this paper aims at reducing $P(c \mid x)$ for unknown samples $x \in \mathcal{D}_u$ by improving the feature representation and network output to be more robust to unknown samples.

We review a few details of the most related approaches to which we compare: thresholding softmax scores, estimating uncertainty, taking an open-set approach, and using a background class.

**Thresholding Softmax Scores:**    This approach assumes that samples from a class on which the network was not trained would have probability scores distributed across all the known classes, hence making the maximum softmax score for any of the known classes low. Therefore, if the system thresholds the maximum score, it may avoid classifying such a sample as one of the known classes. While rejecting unknown inputs by thresholding some type of score is common [24, 7, 9], thresholding softmax is problematic. Almost since its inception [2], softmax has been known to bias the probabilities towards a certain class even though the difference between the logit values of the winner class and its neighboring classes is minimal. This was highlighted by Matan *et al.* [24] who noted that softmax would increase scores for a particular class even though they may have very limited activation on the logit level. In order to train the network to provide better logit values, they included an additional parameter $\alpha$ in the softmax loss by modifying the loss function as: $S_c(x) = \frac{\log e^{l_c(x)}}{e^\alpha + \sum\limits_{c' \in \mathcal{C}} e^{l_{c'}(x)}}$. This modification forces the network to have a higher loss when the logit values $l_c(x)$ are smaller than $\alpha$ during training, and decreases the softmax scores when all logit values are smaller than $\alpha$. This additional parameter can also be interpreted as an additional node in the output layer that is not updated during backpropagation. The authors also viewed this node as a representation of *none of the above*, i.e., the node accounts for $x \in \mathcal{D}_u$.

**Uncertainty Estimation:**    In 2017, Lakshminarayanan *et al.* [20] introduced an approach to predict uncertainty estimates using MLP ensembles trained with MNIST digits and their adversarial examples. Rather than approximating $P(u \mid x)$, their approach is focused at reducing $\max_c P(c \mid x)$ whenever $x \in \mathcal{D}_u$, which they solved using a network ensemble. We compare our approach to their results using their evaluation processes as well as using our Open-Set Classification Rate (OSCR) curve.

**Open-Set Approach OpenMax:**    The OpenMax approach introduced by Bendale and Boult [1] tackles deep networks in a similar way to softmax as it does not use background samples during training, i.e., $\mathcal{D}'_b = \varnothing$. OpenMax aims at directly estimating $P(\mathcal{U} \mid x)$. Using the deep features from training samples, it builds per-class probabilistic models of the input not belonging to the known classes, and combines these in the OpenMax estimate of each class probability, including $P(\mathcal{U} \mid x)$. Though this approach provided the first steps to formally address the open-set issue for deep networks, it is an offline solution after the network had already been trained. It does not improve the feature representation to better detect unknown classes.

**Background Class:**    Interestingly, none of the previous works compared with or combined with the background class modeling approach that dominates state-of-the-art detection approaches, i.e., most of the above approaches assumed $\mathcal{D}_b = \varnothing$. The background class approach can be seen as an extension of the softmax approach by Matan *et al.* [24] seen above. In this variation, the network is trained to find an optimal value of $\alpha$ for each sample such that the resulting plane separates unknown samples from the rest of the classes. Systems trained with a background class use samples from $\mathcal{D}'_b$, hoping that these samples are sufficiently representative of $\mathcal{D}_u$, so that after training the system correctly labels unknown, i.e., they assume $\forall x \in \mathcal{D}'_b \colon P(\mathcal{U} \mid x) \approx 1 \implies \forall z \in \mathcal{D}_a, c \in \mathcal{C} \colon P(\mathcal{U} \mid z) > P(c \mid z)$.

While this is true by construction for most of the academic datasets like PASCAL [8] and MS-COCO [22], where algorithms are often evaluated, it is a likely source of "negative" dataset bias [37] and does not necessarily hold true in the real world where the negative space has near infinite variety of inputs that need to be rejected. To the best of our knowledge, this approach has never been formally tested for open-set effectiveness, i.e., handling unknown unknown samples from $\mathcal{D}_a$.

Though all of these approaches provide partial solutions to address the problem of unknown samples, we show that our novel approach advances the state of the art in open-set image classification.

## 3    Visualizing Deep Feature Responses to Unknown Samples

In order to highlight some of the issues and understand the response of deep networks to out of distribution or unknown samples, we create a visualization of the responses from deep networks while encountering known and unknown samples. We use the LeNet++ network [39], which aims at classifying the samples in the MNIST hand-written digits database [21] while representing each sample in a two dimensional deep feature space. This allows for an easy visualization of the deep feature space which can be seen as an imitation of the response of the network.

We train the network to classify the MNIST digits ($\mathcal{D}_c$) and then feed characters from an unknown dataset ($\mathcal{D}_a$) to obtain the response of the network. In Fig. 1, we sampled unknowns (black points) from the Devanagari[28] dataset, while other plots in the supplemental material use samples from other unknown datasets. As seen in Figure 1(a), when using the standard softmax approach there is quite an overlap between features from $\mathcal{D}_c$ and $\mathcal{D}_a$. Furthermore, from the histogram of the softmax scores it is clear that majority of unknown samples have a high softmax score for one of the known classes. This means that if a probability threshold $\theta$ has to be chosen such that we get a low number of false positives i.e. less unknown samples are identified as a known class, we would also be rejecting most of the known samples since they would be below $\theta$. Clearly, when a network is not explicitly trained to identify unknown samples it can result in significant confusion between known and unknown samples.

The background class approach explicitly trains with out of distribution samples $\mathcal{D}'_b$, while learning to classify $\mathcal{D}_c$. Here, the goal is to account for any unknown inputs $\mathcal{D}_a$ that occur at test time. In Fig. 1(b) we display results from such an approach where during training NIST letters [15] were used as $\mathcal{D}'_b$. It can be observed that majority of the unknown samples $\mathcal{D}_a$, from the Devanagari dataset, fall within the region of the background class. However, there are still many samples from $\mathcal{D}_a$ overlapping the samples from $\mathcal{D}_c$, mostly at the origin where low probabilities are to be expected. Many $\mathcal{D}_a$ samples also overlap with the neighboring known classes far from the origin and, therewith, obtain high prediction probabilities for those known classes.

For our Objectosphere approach, we follow the same training protocol as in the background approach i.e. training with $\mathcal{D}'_b$. Here, we aim at mapping samples from $\mathcal{D}'_b$ to the origin while pushing the lobes representing the MNIST digits $\mathcal{D}_c$ farther from the origin. This results in a much clearer separation between the known $\mathcal{D}_c$ and the unknowns samples $\mathcal{D}_a$, as visible in Fig. 1(c).

## 4    Approach

One of the limitations of training with a separate background class is that the features of all unknown samples are required to be in one region of the feature space. This restriction is independent of the similarity a sample might have to one of the known classes. An important question not addressed in prior work is if there exists a better and simpler representation, especially one that is more effective at creating a separation between known and unknown samples.

From the depiction of the test set of MNIST and Devanagari dataset in Figs. 1(a) and 2(a), we observe that magnitudes for unknown samples in deep feature space are often lower than those of known samples. This observation leads us to believe that the magnitude of the deep feature vector captures information about a sample being unknown. We want to exploit and exaggerate this property to develop a network where for $x \in \mathcal{D}'_b$ we reduce the deep feature magnitude ($\|F(x)\|$) and maximize entropy of the softmax scores in order to separate them from known samples. This allows the network to have unknown samples that share features with known classes as long as they have a small feature magnitude. It might also allow the network to focus learning capacity to respond to the known classes

instead of spending effort in learning specific features for unknown samples. We do this in two stages. First, we introduce the *Entropic Open-Set loss* to maximize entropy of unknown samples by making their softmax responses uniform. Second, we expand this loss into the *Objectosphere loss*, which requires the samples of $\mathcal{D}'_c$ to have a magnitude above a specified minimum while driving the magnitude of the features of samples from $\mathcal{D}'_b$ to zero, providing a margin in both magnitude and entropy between known and unknown samples.

In the following, for classes $c \in \mathcal{C}$ let $S_c(x) = \frac{e^{l_c(x)}}{\sum\limits_{c' \in \mathcal{C}} e^{l_{c'}(x)}}$ be the standard softmax score where $l_c(x)$ represents the logit value for class $c$. Let $F(x)$ be deep feature representation from the fully connected layer that feeds into the logits. For brevity, we do not show the dependency on input $x$ when its obvious.

## 4.1 Entropic Open-Set Loss

In deep networks, the most commonly used loss function is the standard softmax loss given above. While we keep the softmax loss calculation untouched for samples of $\mathcal{D}'_c$, we modify it for training with the samples from $\mathcal{D}'_b$ seeking to equalize their logit values $l_c$, which will result in equal softmax scores $S_c$. The intuition here is that if an input is unknown, we know nothing about what classes it relates to or what features we want it to have and, hence, we want the maximum entropy distribution of uniform probabilities over the known classes. Let $S_c$ be the softmax score as above, our Entropic Open-Set Loss $J_E$ is defined as:

$$J_E(x) = \begin{cases} -\log S_c(x) & \text{if } x \in \mathcal{D}'_c \text{ is from class } c \\ -\frac{1}{C} \sum\limits_{c=1}^{C} \log S_c(x) & \text{if } x \in \mathcal{D}'_b \end{cases} \tag{1}$$

We first show that the minimum of the loss $J_E$ for sample $x \in \mathcal{D}_b$ is achieved when the softmax scores $S_c(x)$ for all known classes are identical.

**Lemma 1.** *For an input $x \in \mathcal{D}'_b$, the loss $J_E(x)$ is minimized when all softmax responses $S_c(x)$ are equal: $\forall c \in \mathcal{C} : S_c(x) = S = \frac{1}{C}$.*

For $x \in \mathcal{D}'_b$, the loss $J_E(x)$ is similar in form to entropy over the per-class softmax scores. Thus, based on Shannon [35], it is intuitive that the loss is minimized when all values are equal. However, since $J_E(x)$ is not exactly identical to entropy, a formal proof is given in the supplementary material.

**Lemma 2.** *When the logit values are equal, the loss $J_E(x)$ is minimized.*

*Proof.* If the logits are equal, say $l_c = \eta$, then each softmax has an equivalent numerator ($e^\eta$) and, hence, all softmax scores are equal. $\square$

**Theorem 1.** *For networks whose logit layer does not have bias terms, and for $x \in \mathcal{D}'_b$, the loss $J_E(x)$ is minimized when the deep feature vector $F(x)$ that feeds into the logit layer is the zero vector, at which point the softmax responses $S_c(x)$ are equal: $\forall c \in \mathcal{C} : S_c(x) = S = \frac{1}{C}$ and the entropy of softmax and the deep feature is maximized.*

*Proof.* Let $F \in \mathbb{R}^M$ be our deep feature vector, and $W_c \in \mathbb{R}^M$ be the weights in the layer that connects $F$ to the logit $l_c$. Since the network does not have bias terms, $l_c = W_c \cdot F$, so when $F = \vec{0}$, then the logits are all equal to zero: $\forall c : l_c = 0$. By Lemma 2, we know that when the logits are all equal the loss $J_E(x)$ is minimized and softmax scores are equal, and maximize entropy. $\square$

Note that the theorem does not show that $F = \vec{0}$ is the only minimum because it is possible that there exists a subspace of the feature space that is orthogonal to all $W_c$. Minimizing loss $J_E(x)$ may, but does not have to, result in a small magnitude on unknown inputs. A small perturbation from such a subspace may quickly increase decrease entropy, so we seek a more stable solution.

## 4.2 Objectosphere Loss

Following the above theorem, the Entropic Open-Set loss produces a network that generally represents the unknown samples with very low magnitudes, while also producing high softmax entropy. This can be seen in Fig. 2(b) where magnitudes of known unknown test samples ($\mathcal{D}_b$) are well-separated from magnitudes of known samples ($\mathcal{D}_c$). However, there is often a modest overlap between the feature

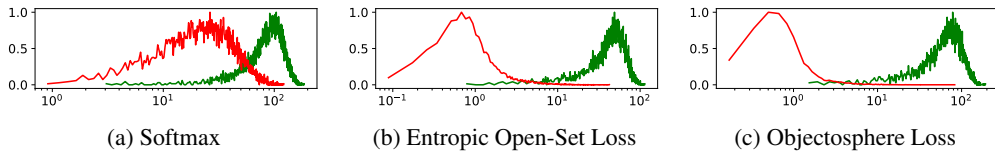

| (a) Softmax | (b) Entropic Open-Set Loss | (c) Objectosphere Loss |

Figure 2: NORMALIZED HISTOGRAMS OF DEEP FEATURE MAGNITUDES. *In (a) the magnitude of the unknown samples ($\mathcal{D}_a$) are generally lower than the magnitudes of the known samples ($\mathcal{D}_c$) for a typical deep network. Using our novel Entropic Open-Set loss (b) we are able to further decrease the magnitudes of unknown samples and using our Objectosphere loss (c) we are able to create an even better separation between known and unknown samples.*

magnitudes of known and unknown samples. This should not be surprising as nothing is forcing known samples to have a large feature magnitude or always force unknown samples to have small feature magnitude. Thus, we attempt to put a distance margin between them. In particular, we seek to push known samples into what we call the *Objectosphere* where they have large feature magnitude and low entropy – we are training the network to have a large response to known classes. Also, we penalize $\|F(x)\|$ for $x \in \mathcal{D}'_b$ to minimize feature length and maximize entropy, with the goal of producing a network that does not highly respond to anything other than the class samples. Targeting the deep feature layer helps to ensure there is no accidental minima. Formally, the Objectosphere loss is calculated as:

$$J_R = J_E + \lambda \begin{cases} \max(\xi - \|F(x)\|, 0)^2 & \text{if } x \in \mathcal{D}'_c \\ \|F(x)\|^2 & \text{if } x \in \mathcal{D}'_b \end{cases} \tag{2}$$

which both penalizes the known classes if their feature magnitude is inside the boundary of the Objectosphere, and unknown classes if their magnitude is greater than zero. We now prove this has only one minimum.

**Theorem 2.** *For networks whose logit layer does not have bias terms, given an known unknown input $x \in \mathcal{D}'_b$, the loss $J_R(x)$ is minimized if and only if the deep feature vector $F = \vec{0}$, which in turn ensures the softmax responses $S_c(x)$ are equal: $\forall c \in \mathcal{C} : S_c(x) = S = \frac{1}{C}$ and maximizes entropy.*

*Proof.* The "if" follows directly from Theorem 1 and the fact that adding 0 does not change the minimum and given $F = \vec{0}$, the logits are zero and the softmax scores must be equal. For the "only if", observe that of all possible features $F$ with $\forall c \in \mathcal{C}: W_c \cdot F = 0$ that minimize $J_E$, the added $\|F(x)\|^2$ ensures that the only minimum is at $F = \vec{0}$. □

The parameter $\xi$ sets the margin, but also implicitly increases scaling and can impact learning rate; in practice one can determine $\xi$ using cross-class validation. Note that larger $\xi$ values will generally scale up deep features, including the unknown samples, but what matters is the overall separation. As seen in the histogram plots of Fig. 2(c), compared to the Entropic Open-Set loss, the Objectosphere loss provides an improved separation in feature magnitudes.

Finally, instead of thresholding just on the final softmax score $S_c(x)$ of our Objectosphere network, we can use the fact that we forced known and unknown samples to have different deep feature magnitudes and multiply the softmax score with the deep feature magnitude: $S_c(x) \cdot \|F(x)\|$. Thresholding this multiplication seems to be more reasonable and justifiable.

### 4.3 Evaluating Open-Set Systems

An open-set system has a two-fold goal, it needs to reject samples belonging to unknown classes $\mathcal{D}_u$ as well as classify the samples from the correct classes $\mathcal{D}_c$. This makes evaluating open-set more complex. Various evaluation metrics attempt to handle the unknwon classes $\mathcal{D}_u$ in their own way but have certain drawbacks which we discuss for each of these measures individually.

**Accuracy v/s Confidence Curve:** In this curve, the accuracy or precision of a given classification network is plotted against the threshold of softmax scores, which are assumed to be confidences. This curve was very recently applied by Lakshminarayanan *et al.* [20] to compare algorithms on their robustness to unknown samples $\mathcal{D}_u$. This measure has the following drawbacks:

1. *Separation between performance on $\mathcal{D}_u$ and $\mathcal{D}_c$:* In order to provide a quantitative value on the vertical axis, i.e., accuracy or precision, samples belonging to the unknown classes $\mathcal{D}_u$ are

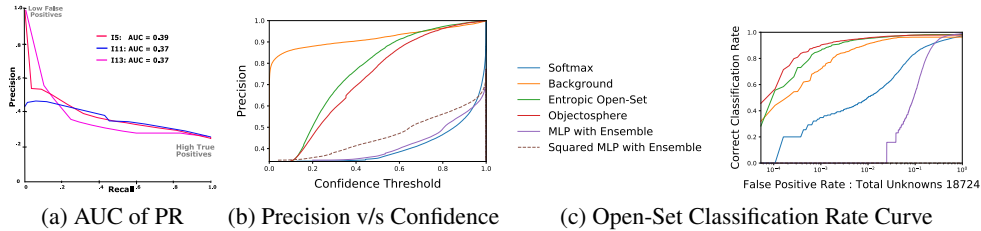

| (a) AUC of PR | (b) Precision v/s Confidence | (c) Open-Set Classification Rate Curve |

Figure 3: COMPARISON OF EVALUATION METRICS. *In (a) we depict the Area under the Curve (AUC) of Precision-Recall Curves applied to the data from the CriteoLabs display ad challenge on Kaggle[2]. The algorithm with the maximum AUC (I5) does not have best performance at almost any recall. In (b), our algorithms are compared to the MLP ensemble of [20] using their accuracy v/s confidence curve using MNIST as known and NotMNIST as unknown samples. In (c), the proposed Open-Set Classification Rate curves are provided for the same algorithms. While MLP and Squared MLP actually come from the same algorithm, they have a different performance in (b), but are identical in (c).*

considered as members of another class which represents all classes of $\mathcal{D}_u$, making the total number of classes for the purpose of this evaluation $C + 1$. Therefore, this measure can be highly sensitive to the dataset bias since the number of samples belonging to $\mathcal{D}_u$ may be many times higher than those belonging to $\mathcal{D}_c$. One may argue that a simplified weighted accuracy might solve this issue but it still would not be able to provide a measure of the number of samples from $\mathcal{D}_u$ that were classified as belonging to a known class $c \in \mathcal{C}$.

2. *Algorithms are Incomparable:* Since confidences across algorithms cannot be meaningfully related, comparing them based on their individual confidences becomes tricky. An algorithm may classify an equal number of unknown samples of $\mathcal{D}_u$ as one of the known classes at a confidence of 0.1 to another algorithm at a confidence of 0.9, but still have the same precision since different number of known samples of $\mathcal{D}_c$ are being classified correctly.

3. *Prone to Scaling Errors:* An ideal evaluation metric should be independent of any monotonic re-normalization of the scores. In Fig. 3(b), we added a curve labeled squared MLP with Ensemble, which appears to be better than the MLP ensemble though it is the same curve with softmax scores scaled by simply squaring them.

**Area Under the Curve (AUC) of a Precision Recall Curve**   AUC is another evaluation metric commonly found in research papers from various fields. In object detection, it is popularly calculated for a precision-recall (PR) curve and referred to as average precision (AP). The application of any algorithm to a real world problem involves the selection of an operating point, with the natural choices on a PR curve being either high precision (low number of false positives) or high recall (high number of true positives). Let us consider the PR curves in Fig. 3(a), which are created from real data. When high precision of 0.8 is chosen as an operating point, the algorithm I13 provides a better recall than I5 but this information is not clear from the AUC measure since I5 has a larger AUC than I13. In fact, even though I11 has same AUC as I13, it can not operate at a precision $> 0.5$. A similar situation exists when selecting an operating point based on the high recall. This clearly depicts that, though the AUC of a PR curve is a widely used measure in the research community, it cannot be reliably used for selecting one algorithm over another. Also other researchers have pointed out that "*AP cannot distinguish between very different [PR] curves*" [27]. Moreover as seen in Fig. 3(a), the PR curves are non-monotonic by default. When object detection systems are evaluated, the PR curves are manually made monotonic by assigning maximum precision value at any given recall for all recalls larger than the current one, which provides an over-optimistic estimation of the final AUC value.

**Recall@K**   According to Plummer *et al.* [30], "*Recall@K (K = 1, 5, 10) [is] the percentage of queries for which a correct match has rank of at most K.*" This measure has the same issue of the separation between performance on $\mathcal{D}_u$ and $\mathcal{D}_c$ as the accuracy v/s confidence curve. Recall@K can only be assumed as an open-set evaluation metric due to the presence of the background class in that paper, i.e., the total number of classes for the purpose of this evaluation is also $C + 1$. Furthermore, in a detection setup the number of region proposals for two algorithms are dependent on their underlying approaches. Therefore, the Recall@K is not comparable for two algorithms since the number of samples being compared are different.

| Algorithm | $\mathcal{D}_c$ Entropy | $\mathcal{D}_a$ Entropy | $\mathcal{D}_c$ Magnitude | $\mathcal{D}_a$ Magnitude |
|---|---|---|---|---|
| Softmax | 0.015± .084 | 0.318± .312 | 94.90± 27.47 | 32.27 ± 18.47 |
| Entropic Open-Set | 0.050± .159 | 1.984± .394 | 50.14± 17.36 | 1.50 ± 2.50 |
| Objectosphere | 0.056± .168 | 2.031± .432 | 76.80± 28.55 | 2.19 ± 4.73 |

Table 1: ENTROPY AND FEATURE MAGNITUDE. *Mean and standard deviation of entropy and feature magnitudes for known and unknown test samples are presented for different algorithms on Experiment #1 (LeNet++). As predicted by the theory, Objectosphere has the highest entropy for unknown samples ($\mathcal{D}_a$) and greatest separation and between known ($\mathcal{D}_c$) and unknown ($\mathcal{D}_a$) for both entropy and deep feature magnitude.*

**Our Evaluation Approach**    To properly address the evaluation of an open-set system, we introduce the *Open-Set Classification Rate* (OSCR) curve as shown in Fig. 3(c), which is an adaptation of the *Detection and Identification Rate* (DIR) curve used in open-set face recognition [29]. For evaluation, we split the test samples into samples from known classes $\mathcal{D}_c$ and samples from unknown classes $\mathcal{D}_u$. Let $\theta$ be a score threshold. For samples from $\mathcal{D}_c$, we calculate the *Correct Classification Rate* (CCR) as the fraction of the samples where the correct class $\hat{c}$ has maximum probability and has a probability greater than $\theta$. We compute the *False Positive Rate* (FPR) as the fraction of samples from $\mathcal{D}_u$ that are classified as *any* known class $c \in \mathcal{C}$ with a probability greater than $\theta$:

$$\text{FPR}(\theta) = \frac{\left|\{x \mid x \in \mathcal{D}_a \wedge \max_c P(c \mid x) \geq \theta\}\right|}{|\mathcal{D}_a|} \,,$$

$$\text{CCR}(\theta) = \frac{\left|\{x \mid x \in \mathcal{D}_c \wedge \arg \max_c P(c \mid x) = \hat{c} \wedge P(\hat{c}|x) > \theta\}\right|}{|\mathcal{D}_c|} \,. \tag{3}$$

Finally, we plot CCR versus FPR, varying the probability threshold large $\theta$ on the left side to small $\theta$ on the right side. For the smallest $\theta$, the CCR is identical to the closed-set classification accuracy on $\mathcal{D}_c$. Unlike the above discussed evaluation measures, which are prone to dataset bias, OSCR is not since its DIR axis is computed solely from samples belonging to $\mathcal{D}_c$. Moreover, when algorithms exposed to different number of samples from $\mathcal{D}_a$ need to be compared, rather than using the normalized FPR with an algorithm specific $\mathcal{D}_a$, we may use the raw number of false positives on the horizontal axis [16].

## 5   Experiments

Our experiments demonstrate the application of our Entropic Open-Set loss and Objectosphere loss while comparing them to the background class and standard softmax thresholding approaches for two types of network architectures. The first set of experiments use a two dimensional LeNet++ architecture for which experiments are detailed in Sec. 3. We also compare our approaches to the recent state of the art OpenMax [1] approach, which we significantly outperform as seen in Fig. 1(d). Our experiments include Devanagari, CIFAR-10 and NotMNIST datasets as $\mathcal{D}_a$, the results being summarized in Tab. 2 with more visualizations in the supplemental material. In Fig. 3(c), we use NotMNIST as $\mathcal{D}_a$ and significantly outperform the adversarially trained MLP ensemble from Lakshminarayanan *et al.* [20].

The second set of experiments use a ResNet-18 architecture with a 1024 feature dimension layer to classify the ten classes from CIFAR-10 [18], $\mathcal{D}_c$. We use the super classes of the CIFAR-100 [18] dataset to create a custom split for our $\mathcal{D}_b$ and $\mathcal{D}_a$ samples. We split the super classes into two equal parts, all the samples from one of these splits are used for training as $\mathcal{D}_b$, while samples from the other split is used only during testing as $\mathcal{D}_a$. Additionally, we also test on the Street View House Numbers (SVHN) [26] dataset as $\mathcal{D}_a$. The results for both the CIFAR-100 and SVHN dataset are summarized in Tab. 2. In addition to the Entropic Open-Set and Objectosphere loss we also test the scaled objectosphere approach mentioned in Sec. 4.2.

## 6   Discussion and Conclusion

The experimental evidence provided in Tab. 1 supports the theory that samples from unseen classes generally have a low feature magnitude and higher softmax entropy. Our Entropic Open-Set loss

Algorithm details: `http://www.kellygwiseman.com/criteo-labs-advertising-challenge`

| Experiment | Unknowns $|\mathcal{D}_a|$ | Algorithm | CCR at FPR of | | | |
|---|---|---|---|---|---|---|
| | | | $10^{-4}$ | $10^{-3}$ | $10^{-2}$ | $10^{-1}$ |
| LeNet++ Architecture Trained with MNIST digits as $\mathcal{D}_c$ and NIST Letters as $\mathcal{D}_b$ | Devanagri 10032 | Softmax | 0.0 | 0.0 | 0.0777 | 0.9007 |
| | | Background | 0.0 | 0.4402 | 0.7527 | 0.9313 |
| | | Entropic Open-Set | 0.7142 | 0.8746 | 0.9580 | 0.9788 |
| | | Objectosphere | **0.7350** | **0.9108** | **0.9658** | **0.9791** |
| | NotMNIST 18724 | Softmax | 0.0 | 0.3397 | 0.4954 | 0.8288 |
| | | Background | 0.3806 | 0.7179 | 0.9068 | 0.9624 |
| | | Entropic Open-Set | 0.4201 | 0.8578 | 0.9515 | **0.9780** |
| | | Objectosphere | **0.512** | **0.8965** | **0.9563** | 0.9773 |
| | CIFAR10 10000 | Softmax | 0.7684 | 0.8617 | 0.9288 | 0.9641 |
| | | Background | 0.8232 | 0.9546 | 0.9726 | 0.973 |
| | | Entropic Open-Set | **0.973** | **0.9787** | **0.9804** | **0.9806** |
| | | Objectosphere | 0.9656 | 0.9735 | 0.9785 | 0.9794 |
| ResNet-18 Architecture Trained with CIFAR-10 Classes as $\mathcal{D}_c$ and Subset of CIFAR-100 as $\mathcal{D}_b$ | SVHN 26032 | Softmax | 0.1924 | 0.2949 | 0.4599 | 0.6473 |
| | | Background | 0.2012 | 0.3022 | 0.4803 | 0.6981 |
| | | Entropic Open-Set | 0.1071 | 0.2338 | 0.4277 | 0.6214 |
| | | Objectosphere | 0.1862 | 0.3387 | 0.5074 | 0.6886 |
| | | Scaled Objecto | **0.2547** | **0.3896** | **0.5454** | **0.7013** |
| | CIFAR-100 Subset 4500 | Softmax | N/A | 0.0706 | 0.2339 | 0.5139 |
| | | Background | N/A | 0.1598 | 0.3429 | 0.6049 |
| | | Entropic Open-Set | N/A | 0.1776 | 0.3501 | 0.5855 |
| | | Objectosphere | N/A | 0.1866 | 0.3595 | 0.6345 |
| | | Scaled Objecto | N/A | **0.2584** | **0.4334** | **0.6647** |

Table 2: EXPERIMENTAL RESULTS. *Correct Classification Rates (CCR) at different False Positive Rates (FPR) are given for multiple algorithms tested on different datasets. For each experiment and at each FPR, the best performance is in bold. We show Scaled Objectosphere only when it was better than Objectosphere; magnitude scaling does not help in the 2D feature space of LeNet++.*

and Objectosphere loss utilize this default behaviour by further increasing entropy and decreasing magnitude for unknown inputs. This improves network robustness towards out of distribution samples as supported by our experimental results in Tab. 2. Here, we summarize results of the Open-Set Classification Rate (OSCR) Curve by providing the Correct Classification Rates (CCR) at various False Positive Rate (FPR) values.

The proposed solutions are not, however, without their limitations which we now discuss. Though training with the Entropic Open-Set loss is at about the same complexity as training with a background class, the additional magnitude restriction for the Objectosphere loss can make the network a bit more complicated to train. The Objectosphere loss requires determining $\lambda$, which is used to balance two elements of the loss, as well as choosing $\xi$, the minimum feature magnitude for known samples. These can be chosen systematically, using cross-class calibration, where one trains with a subset of the background classes, say half of them, then tests on the remaining unseen background classes. However, this adds complexity and computational cost.

We also observe that in case of the LeNet++ architecture, some random initializations during training result in the Entropic Open-Set loss having better or equivalent performance to Objectosphere loss. This may be attributed to the narrow two dimensional feature space used by the network. In high dimensional feature space as in the ResNet-18 architecture the background class performs better than Entropic Open-Set and Objectosphere at very low FPR, but is beat by the Scaled-Objectosphere approach, highlighting the importance of low feature magnitudes for unknowns.

During our experiments, it was also found that the choice of unknown samples used during training is important. E.g. in the LeNet++ experiment, training with CIFAR samples as the unknowns $\mathcal{D}'_b$ does not provide robustness to unknowns from the samples of NIST Letters dataset, $\mathcal{D}_a$. Whereas, training with NIST Letters $\mathcal{D}'_b$ does provide robustness against CIFAR images $\mathcal{D}_a$. This is because CIFAR images are distinctly different from the MNIST digits where as NIST letters have attributes very similar to them. This finding is consistent with the well known importance of hard-negatives in deep network training.

While there was considerable prior work on using a background or garbage class in detection, as well as work on open set, rejection, out-of-distribution detection, or uncertainty estimation, this paper

presents the first theoretically grounded significant steps to an improved network representation to address unknown classes and, hence, reduce network agnostophobia.

## 7    Acknowledgements

This research is based upon work funded in part by NSF IIS-1320956 and in part by the Office of the Director of National Intelligence (ODNI), Intelligence Advanced Research Projects Activity (IARPA), via IARPA R&D Contract No. 2014-14071600012. The views and conclusions contained herein are those of the authors and should not be interpreted as necessarily representing the official policies or endorsements, either expressed or implied, of the ODNI, IARPA, or the U.S. Government. The U.S. Government is authorized to reproduce and distribute reprints for Governmental purposes notwithstanding any copyright annotation thereon.

## Footnotes

[1] http://github.com/Vastlab/Reducing-Network-Agnostophobia

[2]Challenge website: `http://www.kaggle.com/c/criteo-display-ad-challenge`

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
