[Supplementary Material · 5501_supplementary.pdf]

## Supplementary Material: Reducing Network Agnostophobia

### Proof of Lemma 1

### Lemma 1

For an input $x \in \mathcal{D}_b'$, Entropic Open-Set loss $J_E(x)$ is minimized when all softmax responses $S_c(x)$ are equal: $\forall c \in \{1, \ldots, C\} : S_c(x) = S = \frac{1}{C}$.

*Proof.* Assume that, at the minimum, softmax responses are not equal. We can represent any small deviation from equality as $\exists \{\delta_c \mid c \in \{1, \ldots, K\} \wedge 0 < \delta_c \leq S\}$ with $K < C$ such that $S_c = S + \delta_c$. Simultaneously, $\exists \{\delta_{c'} \mid c' \in \{K+1, \ldots, C\} \wedge 0 \leq \delta_{c'} \leq S\}$ with $S_{c'} = S - \delta_{c'}$. Note that $\delta_{c'}$ cannot be larger than $S$ since $S_{c'}$ cannot be smaller than 0. Finally, $\sum_{c=1}^{K} \delta_c = \sum_{c'=K+1}^{C} \delta_{c'}$. If this reduces the loss $J_E(x)$ then:

$$-\left( \sum_{c=1}^{K} \log\left(S + \delta_c\right) + \sum_{c'=K+1}^{C} \log(S - \delta_{c'}) \right) < -\sum_{c=1}^{C} \log S_c \,.$$

With $\log(S + \delta) = \log(S \cdot (1 + \delta/S)) = \log(S) + \log(1 + \delta/S)$, we have:

$$\sum_{c=1}^{K} \left( \log S + \log\left(1 + \frac{\delta_c}{S}\right) \right) + \sum_{c'=K+1}^{C} \left( \log S + \log\left(1 - \frac{\delta_{c'}}{S}\right) \right) > \sum_{c=1}^{C} \log S_c$$

$$\sum_{c=1}^{K} \log\left(1 + \frac{\delta_c}{S}\right) + \sum_{c'=K+1}^{C} \log\left(1 - \frac{\delta_j}{S}\right) + \sum_{c=1}^{C} \log S_c > \sum_{c=1}^{C} \log S_c$$

$$\sum_{c=1}^{K} \log\left(1 + \frac{\delta_c}{S}\right) + \sum_{c'=K+1}^{C} \log\left(1 - \frac{\delta_{c'}}{S}\right) > 0$$

Using the Taylor series $\log(1 + x) = \sum_{n=1}^{\infty} \frac{(-1)^{n-1} x^n}{n}$ and $\log(1 - x) = -\sum_{n=1}^{\infty} \frac{x^n}{n}$, we get:

$$\sum_{c=1}^{K} \left( \sum_{n=1}^{\infty} \frac{(-1)^{n-1} \delta_c^n}{nS^n} \right) - \sum_{c'=K+1}^{C} \left( \sum_{n=1}^{\infty} \frac{\delta_{c'}^n}{nS^n} \right) > 0$$

$$\sum_{c=1}^{K} \left( \frac{\delta_c}{S} + \sum_{n=2}^{\infty} \frac{(-1)^{n-1} \delta_c^n}{nS^n} \right) - \sum_{c'=k+1}^{C} \left( \frac{\delta_{c'}}{S} + \sum_{n=2}^{\infty} \frac{\delta_{c'}^n}{nS^n} \right) > 0$$

Rearranging and using the constraint that $\sum_{c=1}^{K} \delta_c = \sum_{c'=K+1}^{C} \delta_{c'}$, we obtain:

$$\sum_{c=1}^{K} \frac{\delta_c}{S} - \sum_{c'=K+1}^{C} \frac{\delta_{c'}}{S} + \sum_{c=1}^{K} \sum_{n=2}^{\infty} \frac{(-1)^{n-1} \delta_c^n}{nS^n} - \sum_{c'=K+1}^{C} \sum_{n=2}^{\infty} \frac{\delta_{c'}^n}{nS^n} > 0$$

This yields a contradiction as we assumed $\forall c \in \{1, \ldots, C\} : 0 < \delta_c \leq S$. The right sum is subtracted and is always $< 0$. The left sum yields $-\delta_c^2/(2S^2) + \delta_c^3/(3S^3) - \ldots$ which is strictly smaller than 0 for $0 < \delta_c \leq S$. Thus for unknown inputs, the loss is minimized when all softmax outputs have equal value. $\square$

**Response to Known Unknown Samples $\mathcal{D}_b$: Letters**

(a) Softmax  (b) Background  (c) Objectosphere  (d) OpenSet Recognition Curve

Figure 4: LeNet++ Responses To Known Unknowns. *This figure shows responses of a network trained to classify MNIST digits and reject Latin letters when exposed to samples of classes that the network was trained on.*

**Response to Unknown Unknown Samples $\mathcal{D}_a$: CIFAR**

(a) Softmax  (b) Background  (c) Objectosphere  (d) OpenSet Recognition Curve

Figure 5: LeNet++ Responses To CIFAR Images. *This figure shows responses of a network trained to classify MNIST digits and reject Latin letters when exposed to samples from CIFAR dataset, which are very different from both the classes of interest (MNIST) and the background classes (Letters).*

**Response to Unknown Unknown Samples $\mathcal{D}_a$: NotMNIST**

(a) Softmax  (b) Background  (c) Objectosphere  (d) OpenSet Recognition Curve

Figure 6: LeNet++ Responses to Characters from Not MNIST dataset. *This figure shows responses of a network trained to classify MNIST digits and reject Latin letters when exposed to characters from the Not MNIST dataset.*