[Reviews · NeurIPS 2018]

Reviewer 1



This work proposes an alternative to the softmax loss with the intention of maximizing entropy for unknown categories. This is achieved by explicitly enforcing uniform class logits for unknown classes with the entropic open-set loss, and is further improved by reducing feature magnitude for unknown classes. +Empirical results in Table 2 suggest it can be a better choice of loss than the typical softmax, given the added complexity and proper tuning of \lambda. +Paper was clearly explained and easy to understand +Analysis does a reasonable job of explaining some limitations -Explanation for why new evaluation metrics are introduced appears incomplete. While confidences cannot be directly compared across different detection algorithms, existing works compare the normalized AUC of the precision-recall curve (average precision), which, at the very least, is not affected by monotonic re-normalizations as in the case of the "squared MLP with Ensemble" example. Furthermore, open-set detection tasks such as phrase localization [1] use metrics such as Recall@K. I think it makes sense to include some discussion of why (or when) one should use the proposed evaluation over AUC and Recall@K. [1] Bryan A. Plummer, Liwei Wang, Christopher M. Cervantes, Juan C. Caicedo, Julia Hockenmaier, and Svetlana Lazebnik, Flickr30k Entities: Collecting Region-to-Phrase Correspondences for Richer Image-to-Sentence Models, ICCV, 2015. Post-Rebuttal: The authors have adequately addressed my concerns. I hope they include the comparison with standard evaluations metrics in their final paper.

Reviewer 2



Before putting my review, I would like to underline that I am not an expert in this particular area of NN. It might be that I miss some important related work. The paper presents an approach to let push the losses of networks for unknown samples down. It is done by minimizing the deep features magnitude and maximizing the entropy of the softmax scores for samples from a given background class during training. Openset and Objectosphere loss also increase separation in deep feature magnitude between known and unknown classes. The methods work better than simple thresholding or softmax losses. I think that the paper presents a very nice view on the outlier problem and has good ways to formalize the ideas and visualize the behavior. The only issue I see is the naming of the methods and especially the title of the paper. I don't agree that the network fears the unknown and that this is removed by the methods. It could be maybe even the opposite. Since the losses are lower it could mean that the network is more afraid of the unknown and that it would be good to fear the unknown samples. Maybe the term "Agnostophobia" is already used in the literature (I did a search and did not find it) - still, I suggest being very careful when introducing these "catchy" terms and maybe not do it at all. Minor grammar mistakes should be corrected.

Reviewer 3



# 5501 # Update based on the Authors Response Thank you for the clarification. Based on the response I am willing to vote for an accept. # Summary This paper proposes a new approach to improve the model’s robustness against unknown classes. The authors propose a new objective function that, unlike common detection algorithms, does not add the background class to softmax. Instead, for a given background example, the objective function maximizes the entropy of the softmax layer, which leads to a low magnitude feature representation for the background sample. In addition, the authors propose the use of objectosphere loss to distinguish the known/unknown classes further by penalizing the feature representation of the background sample so that the network does not respond to unknown objects. Theoretical and empirical proof demonstrate the potential capability of the proposed approach. # Quality The authors define the problem well and provide a thorough literature review. The theoretical background is also well explained in the method section. Experimentation also shows the capability of the proposed approach. I would want to see one more set of experiment, in which the proposed approach is used within a real detection pipeline such as, as described in the paper RPN of F-RCNN. # Clarity This is a well-written paper. It flows well, and easy to navigate. It is not explicitly stated in the paper whether the author will release their implementation for the reproducibility of the paper. # Significance The problem defined in the paper is significant especially within the real-world applications of object detection algorithms due to excessive exposure to unknown classes in real-world scenarios. The paper’s solution is significant to tackle these issues.